# Hybrid Deep Learning Models with Sparse Enhancement Technique for Detection of Newly Grown Tree Leaves

**DOI:** 10.3390/s21062077

**Published:** 2021-03-16

**Authors:** Shih-Yu Chen, Chinsu Lin, Guan-Jie Li, Yu-Chun Hsu, Keng-Hao Liu

**Affiliations:** 1Department of Computer Science and Information Engineering, National Yunlin University of Science and Technology, Yunlin 64002, Taiwan; sychen@yuntech.edu.tw (S.-Y.C.); m10717003@gemail.yuntech.edu.tw (G.-J.L.); m10817021@yuntech.edu.tw (Y.-C.H.); 2Artificial Intelligence Recognition Industry Service Research Center, National Yunlin University of Science and Technology, Yunlin 64002, Taiwan; 3Department of Forestry and Natural Resources, National Chiayi University, Chiayi 600355, Taiwan; 4Department of Mechanical and Electro-mechanical Engineering, National Sun Yat-sen University, Kaohsiung 80424, Taiwan; keng3@mail.nsysu.edu.tw

**Keywords:** deep learning, semantic segmentation, imbalanced data, robust principal component analysis, remote sensing image

## Abstract

The life cycle of leaves, from sprout to senescence, is the phenomenon of regular changes such as budding, branching, leaf spreading, flowering, fruiting, leaf fall, and dormancy due to seasonal climate changes. It is the effect of temperature and moisture in the life cycle on physiological changes, so the detection of newly grown leaves (NGL) is helpful for the estimation of tree growth and even climate change. This study focused on the detection of NGL based on deep learning convolutional neural network (CNN) models with sparse enhancement (SE). As the NGL areas found in forest images have similar sparse characteristics, we used a sparse image to enhance the signal of the NGL. The difference between the NGL and the background could be further improved. We then proposed hybrid CNN models that combined U-net and SegNet features to perform image segmentation. As the NGL in the image were relatively small and tiny targets, in terms of data characteristics, they also belonged to the problem of imbalanced data. Therefore, this paper further proposed 3-Layer SegNet, 3-Layer U-SegNet, 2-Layer U-SegNet, and 2-Layer Conv-U-SegNet architectures to reduce the pooling degree of traditional semantic segmentation models, and used a loss function to increase the weight of the NGL. According to the experimental results, our proposed algorithms were indeed helpful for the image segmentation of NGL and could achieve better kappa results by 0.743.

## 1. Introduction

The key to the sustainable development of forest ecosystem resources is to protect the coverage of wild trees. Annual coverage is tracked continuously to relieve the influence of global warming or climate change. According to the forest resource assessment report of UNFAO (United Nations Food and Agricultural Organization), the change in the area of woods, forest biomass, carbon storage, and the improvement of healthy forests are periodic evaluation indexes used for sustainable global forest resources [1]. Therefore, using telemetry to monitor the health standards of forest ecosystems has an important effect on controlling global warming. Climatic change may influence climate events such as the budding and senescence of leaves [2]. Once trees perceive the signal to begin growing in early spring, treetop leaves begin to sprout, newly grown leaves (NGL) develop gradually, and growth of the crown diameter, height, diameter, and carbon storage is further increased [3]. Therefore, NGL can be regarded as the primary key to the response of trees to temperature change and providing key information for early detection of climate change [4]. In forestry, telemetry has been used to investigate species classification [5,6], tree delimitation [7,8], and biomass productivity evaluation [9,10]. However, NGL may have a low probability of occurrence in detection, or they may be smaller than the size of the background. For example, NGL in the forest canopy or NGL in an injured crown in the forest canopy may not be detected effectively by traditional spatial domain image processing techniques [11,12,13,14]. In recent years, unmanned aerial vehicle (UAV) has become an important tool for crop inspection, environmental surveillance, and the detection of various earth surfaces. UAV helps in monitoring different environmental changes [15,16,17], including climate change, ecosystem change, urbanization, and habitat variation. Therefore, the goal of this paper is to investigate robust deep learning-based algorithms to diagnose the growth of trees, and even climatic change by detecting newly grown leaves (NGL) from UAV bitmap images.

Recent years have shown rapid development of deep learning techniques [18] in numerous perceptual tasks like object detection and image understanding, such as genes identification [19,20,21,22,23], cancer diagnosis [24,25], dental caries [26], biomedical/medical imaging [27,28,29], skin lesion [30,31], and crack detection [32,33]. In remote sensing applications, several deep learning frameworks have been presented. For example, in Reference [34], a deep convolutional encoder–decoder model for remote sensing image segmentation was proposed. The model applies the encoder network to extract high-level semantic features of ultra-high-resolution images and uses the decoder network to map the low-resolution encoder feature maps, and creates full input resolution feature maps for pixel-wise labeling. An end-to-end fully convolutional neural network (FCN) [35] was proposed in Reference [36] for the semantic segmentation of remote sensing imaging. An object-based Markov random field (MRF) model with auxiliary label fields was presented in Reference [37] to capture more macro and detailed information for the semantic segmentation of high spatial resolution remote sensing images. Furthermore, symmetrical dense-shortcut deep FCNs for the semantic segmentation of very-high-resolution remote sensing images were proposed in Reference [38]. On the other hand, semantic labeling of 3D urban models over complex scenarios was presented in Reference [39] based on convolutional neural network (CNN) and structure-from-motion techniques. The deep learning method used in this paper belongs to the field of semantic segmentation [35]. The method of semantic segmentation refers to classifying each pixel in an image [40]. The classification results presented by semantic segmentation have more spatial information than traditional classification models and they play an important role in many practical applications. For example, Hyeonwoo Noh et al. [41] discussed image understanding based on semantic segmentation, and Michael Treml et al. [42] investigated automatic driving based on semantic segmentation. In recent years, many related deep learning techniques have been intensively applied to research of leaves such as leaf recognition [43], plant classification [44,45], leaf disease classification, and detection [46,47,48,49,50,51]. Plant and leaf identification [52,53,54] using deep learning is a relatively new field. For example, Reference [43] used a basic and Pre-trained GoogleNet for leaf recognition. Reference [52] applied a CNN model to identify leaf veins. Reference [53] proposed a weed segmentation system by combing the segmentation algorithm and CNN models. Reference [54] proposed LeafNet, a CNN-based plant identification system. The LeafNet architecture is also similar to well-known AlexNet and CifarNet. According to the literature review in this section, there is no research that investigates the effect of the segmentation of hybrid deep learning manners for detection of a newly grown leaf (NGL) in detail. We believe our study first attempted to examine the detection of a newly grown leaf (NGL) using deep learning technique in high-resolution UAV images.

The NGL images used in this article are taken by the UAV drone with a pixel resolution of six centimeters; however, NGL can be smaller than six centimeters. From the image point of view, an NGL is smaller than one pixel (known as sub-pixels). If the popular deep learning models (SegNet [40,55,56,57], LF-SegNet [58], U-Net [59,60,61,62,63,64], U-SegNet [65], etc.) are used directly, some target information will be lost due to the excessive number of pooling performed by the original methods. Therefore, we have added the skip-connection method to connect to the network layer in SegNet. Since the proportion of NGL in the overall image is small (which is an imbalanced data problem in machine learning), we have also adjusted the loss function to improve the detection rate of NGL. On the other hand, it would be helpful for target detection in an image to enhance possible target regions in advance using preprocessing techniques, such as robust principal component analysis (RPCA) or related matrix decomposition techniques based on low-rank and sparse properties [66]. RPCA or its related techniques have been successfully used in several applications of remote sensing or image processing. For example, in Reference [67], a joint reconstruction and anomaly detection framework was presented for compressive hyperspectral images based on Mahalanobis distance-regularized tensor RPCA. An edge-preserving rain removal technique was proposed in Reference [68] for light field image data based on RPCA. Moreover, the problem of background subtraction was formulated and solved by relying on a low-rank matrix completion framework in Reference [69]. Furthermore, an efficient rank-revealing decomposition framework based on randomization was presented in Reference [70] for reconstructions of low-rank and sparse matrices. More researches about RPCA or sparse representation can be found in References [71,72,73,74,75]. In this paper, we have proposed a sparse enhancement (SE) technique to strengthen possible targets in an image before applying the learned deep network for the detection of newly grown tree leaves since based on the characteristic of NGL, it can be considered as a sparse matrix in a remote sensing forest image.

In prior studies about NGL, Lin et al. [4] first investigated the hyperspectral target detection algorithms for the possibility of detecting NGL by using spectral information in active and passive manners. However, target detection algorithms were sensitive to the desired target in the final results. After that, Chen et al. [76] proposed the Optimal Signature Generation Process (OSGP) and adaptive window-based Constrained Energy Minimization (CEM) [77], which could provide steadier detection results and reduce the occurrence of false alarms. Another method is based on data preprocessing, in which the target of detection is enhanced by data preprocessing. Chen et al. [78] used preprocessing to enhance the signal of NGL and adopted the Weighted CEM to further reduce the misrecognition rate in the detection results. Differing from prior documents in terms of target detection point of view, this paper started with the segmentation of deep learning application which achieves pixel-level prediction by classifying each pixel according to its category and dividing the image into foreground and background for binary classification and target detection as previous studies. This paper combined hybrid models with U-NET and SegNet, reducing the number of pooling layers to keep the information of NGL and using skip connection to extract feature information from low-level information in the hopes of enhancing NGL features. Moreover, this paper used sparse enhancement preprocessing before the network architecture of deep learning. As the area of NGL to be detected in the forest image had sparse-like characteristics compared to the full image, the NGL in the original image was enhanced by this method. So, the goal of this paper is to investigate the possibility of hybrid deep learning architectures for the detection of NGL from UAV images and the performance of using sparse enhancement (SE) technique as preprocessing in our proposed models.

## 2. Materials and Methods

In this section, we describe the study site and the dataset in Section 2.1, followed by Section 2.2 and Section 2.3, where we describe the current widely-used CNN models and loss function. Finally, in Section 2.3, Section 2.4, Section 2.5 and Section 2.6, we present the proposed hybrid CNN architecture.

### 2.1. Description of the Study Site

In 2002, there were 17 species of trees planted in Taiwan and the area of hardwood forest was 188.59 hectares [4]. The 17 species included the tropical species “Swietenia macrophylla” in the south of Taiwan, which experiences defoliation for one to two weeks during the middle to late March. New leaves grow rapidly within one to two days and are observable in the air. This study used an eBee Real-Time Kinematic (RTK) drone (developed by SenseFly, Switzerland) carrying a Canon PowerShot S110 camera flying at an altitude of 239.2 ft of the area on 12 July 2014 under very good weather condition. The weather is generally hot and humid. The mean annual temperature is 24.38 °C. Leaf development is a process of dynamic plant growth responding to plant physiology and environmental signals [79]. Leaves start to develop from the apical meristems of branches. It is unachievable to visualize a newly sprouted leaflet from the UAV at the initiation stage of the life cycle of a leaf. However, after a period of gradual development, the newly grown leaf is normally at a size of a centimeter and can be visualized from a distance. July is a good time to fly a UAV to detect newly grown leaves in the south of Taiwan. The spatial resolution of the ground sample distance of the images was 6.75 cm (known as a centimeter-level very high resolution (VHR) image). The research site was located in the Baihe District of Tainan City, Taiwan (23°20′ N, 120°27′ E) as shown in Figure 1. This image includes the red, green, and blue bands, with a dimension of 1000 × 1300 pixels and the actual area of the full image is 60 m × 78 m.

#### Ground Truth

In 2008, a few permanent plots were deployed over the broadleaf forest for research of forest growth [80,81], where a series of ground inventory is annually conducted for stand dynamics [17]. The data have been successfully derived using the forest canopy height model [82] and have investigated the feasibility of global/local target detection algorithms for the detection of NGL by our prior studies [4,76,78]. As shown in Figure 2a,b, the NGLs can be visually inspected by experts due to their appearance. They are bright, light green, and amassed over the tree crowns. According to a row of several years of inventory, the ground truth of the NGL over the images was visually interpreted by two professors from the Department of Forestry at National Chiayi University, Taiwan, and was also validated in situ. To quantify and compare the effects of different algorithms, there must be an NGL detection map as the ground truth of Area 1 and Area 2, as shown in Figure 2c,d. The NGL is only about 3–4% of the entire images.

### 2.2. Deep Learning Models

#### 2.2.1. SegNet

The SegNet network architecture was proposed by the research team of Vijay and Alex at the University of Cambridge [55]. The architecture is divided into a convolutional encoding layer, convolutional layers corresponding to the max-pooling layer, and an up-sampling layer to replace the original max-pooling layer. The reduced feature map is restored to the size of an input image. Finally, the softmax classification layer is applied. The architecture is shown in Figure 3 (there are 26 convolutional layers, five max-pooling layers, and five up-sampling layers). For an image imported into SegNet (if the image size is W × H), the feature map is obtained through the convolutional layers and standardized operation, and the feature map is reduced by the max-pooling layer to a certain scale for the upsampling operation. The feature map is deconvolved to the size of the original input image, therefore, numerous feature maps that are as large as the original image can be obtained. The result is imported into the softmax classifier to analyze the class of each pixel. This paper performed modification using SegNet as the fundamental model, and the pooling frequency of SegNet was reduced to maintain the information of tiny targets.

#### 2.2.2. U-Net

U-Net [60] is a segmentation network proposed by Olaf Ronneberger et al. at the ISBI(International Symposium on Biomedical Imaging) Challenge in 2015 (they won the championship). In medical images, Zongwei Zhou et al. [83] proposed the deep supervision model extended from U-Net for training. Moreover, Ozan Oktay et al. [84] used a new AG model in U-Net, in which the sensitivity and precision of prediction were enhanced. The U-shaped network structure is shown in Figure 4. The overall network structure is similar to SegNet as it uses the skip connection technique to compensate for the spatial information lost after up-sampling of the bottom layer.

### 2.3. Loss Function of the Convolutional Neural Network

In machine learning, the loss function is used to estimate the extent of the error in a model. It is calculated according to the prediction value and the true value. A smaller error indicates the model has a certain recognition capability. The NGL detection classification method in this paper could be regarded as a dichotomy, as the loss function is mostly classified by Binary cross-entropy [85]. However, the NGL accounts for only a small part of the overall data, meaning it is an imbalanced data problem in machine learning. To enable the model to detect as many NGL as possible, we have used the following loss function method with adjustable data weights.

#### 2.3.1. Binary Cross-Entropy (Binary CE or CE)

In deep learning, the common loss function for a binary problem is CE [85], which is expressed as follows:(1)$$\mathrm{Binary}\;\mathrm{CE}\left(p,\;\hat{p}\right)=-\left(\mathrm{plog}\left(\hat{p}\right)+\left(1-p\right)\log\left(1-\hat{p}\right)\right)$$
where p is the ground-truth and $\hat{p}$ is the result predicted for the NGL category. The log loss of all samples represents the average of each sample loss, while the loss for a perfect classification model is 0. However, for the two categories of equal importance, if the sample quantity is extremely unbalanced, it is likely to be ignored when calculating the loss for a small number of samples.

#### 2.3.2. Weighted Cross-Entropy (WCE)

For the problem of an uneven sample number, Aurelio et al. [86] proposed a new method, known as weighted cross-entropy (WCE). WCE imports hyperparameter β to increase the weight of the NGL sample, and the range of β is (1,2). The equation is expressed as follows:(2)$$\mathrm{WCE}\left(p,\;\hat{p}\right)=-\left(\beta \;\mathrm{plog}(\hat{p})+\;(1\;-\;p)\log(1\;-\;\hat{p})\right)$$

#### 2.3.3. Balanced Cross-Entropy (BCE)

The loss function used by Shiwen Pan et al. [87] and us is the BCE method. It is different from WCE because the weight of the NGL sample in calculating the loss is increased and the weight of non-NGL samples is suppressed, as expressed in Equation (3):(3)$$\mathrm{BCE}\left(p,\;\hat{p}\right)=-\left(\beta p\log\left(\hat{p}\right)+\left(1-\beta \right)\left(1-p\right)\log\left(1-\hat{p}\right)\right)$$

### 2.4. Hybrid Convolutional Neural Networks

The classical SegNet model was used as the main skeleton of the model. The original SegNet model has more pooling times. The low-level features of tiny targets will blur or even disappear. Therefore, the number of pooling layers was reduced at first. To increase the detection rate, the model could be downsized on the other hand and the loss of tiny targets could be reduced while maintaining the accuracy. When the skip connection (SC) of U-Net was used, the spatial information of the same level was connected upward in the up-sampling process on the bottom layer, and then convolution of the up-sampling was performed. Finally, the batch normalization was added to each convolutional layer [88] to guarantee data stability.

This paper proposed four models based on SegNet and the Skip connection method. Table 1 describes the codes used in the network architecture diagram, wherein C (the convolution layer) uses the unified parameter, Kernel size: (3,3) the 3 by 3 kernel map is used for convolution, padding: same a layer of 0 is added to the outer ring of diagram before convolution, Activation: ReLU let the data be larger than 0, C64 and C128 represent 64 and 128 masks, the kernel size of the outer layer is (1,1). Moreover, the activation used the sigmoid function, in which the data were set as 0~1 for handling binary problems. The complete architecture is introduced in the next section.

#### 2.4.1. 3-Layer SegNet (3L-SN)

Firstly, to maintain the information of tiny targets, the pooling depth in the first extended model which is based on the SegNet model, is reduced from five layers to three layers. Thus, the information of tiny targets would not be lost for pooling, and the number of weights to be trained for the whole network could be reduced. Figure 5 shows the 3-Layer SegNet architecture after simplifying SegNet, in which only three layers of convolution are maintained. Batch normalization was performed after each convolution to ensure the data would not encounter the gradient problem and to prevent the next layer of activation function from being out of action or saturated.

#### 2.4.2. 3-Layer U-SegNet (3L-USN)

Afterward, based on the 3-Layer SegNet network model, the skip connection (SC) of U-Net was imported. This concept was proposed by P. Kumar et al. [65] and Daimary D. et al. [89]. This paper differs in that the skip connection was applied after SegNet pooling depth was reduced by two layers. The purpose of this architecture was to enhance the spatial information of the sample in the course of up-sampling, to enhance the detection capability. Figure 6 shows the 3-Layer skip-connection SegNet network architecture.

#### 2.4.3. 2-Layer U-SegNet (2L-U-SN)

Figure 7 shows that one more pooling layer was reduced while maintaining the skip connection following the network model in the previous section, and the number of convolutions of the convolutional layer was regulated. The purpose was to validate whether there was still a certain detection capability for NGL under a two-layer pooling degree.

#### 2.4.4. 2-Layer Conv-U-SegNet (2L-Conv-USN)

Figure 8 shows the bottom layer combined with a set of a convolutional layer and BN layer in the encoding stage based on the 2L-U-SN network model, which was used to extract feature information from low-level information in the hopes of enhancing the detection efficiency.

### 2.5. Robust Principal Component Analysis (RPCA) and Sparse Enhancement (SE)

To enhance the efficiency of the deep learning network for NGL detection, this study proposed a processing method based on Robust Principal Component Analysis (RPCA) [67,68,69,90,91,92], and to perform image enhancement for the probable NGL areas in the forest image in advance. The SegNet deep learning network was trained and tested by an enhanced image. RPCA could separate the sparse matrix and low-rank matrix from the input image (regarded as a matrix). RPCA is usually used in image recognition and dimension reduction based on high dimensional data, and it performs linear conversion of the original data by calculating the maximum feature vector. The general data matrix contains the main structural information and probable slight noise information. RPCA decomposes the input matrix into two matrices which are added up; one is a low-rank matrix (including the main structural information) in which the columns or rows are linearly correlated. The other is a sparse matrix (including noise information), expressed as Equation (4):(4)$$\underset{L,S}{\min}\;\mathrm{rank}(L)\;+\;\gamma {\left|\left|S\right|\right|}_{0},\;\mathrm{subject}\;\mathrm{to}\;P\;=\;L\;+\;S,$$
where P is the original data (original input image), L is the low-rank matrix, rank (L) is used for calculating the rank of matrix L, S is the sparse matrix, ${\left|\left|S\right|\right|}_{0}$ is the $l_{0}$-norm of S (number of nonzero coefficients in S), and γ is the regulation parameter. As the operation of l_0-norm is relatively complex, Equation (4) has been proved convertible to Equation (5):(5)$$\underset{L,S}{\min}\mathrm{rank}\left(L\right)+\lambda {\left|\left|S\right|\right|}_{1},\;\mathrm{subject}\;\mathrm{to}\;P\;=\;L\;+\;S,\;$$
wherein λ is the regularization parameter and ${\left|\left|S\right|\right|}_{1}$ is the $l_{1}$-norm of S (the sum of the absolute values of the coefficients in S). The input image P can be decomposed into low-rank matrix L and sparse matrix S by optimizing Equation (2), wherein L is the principal component of the image and S is the image noise with sparsity. In this study as the NGL area to be detected in the forest image had sparsity-like characteristics compared with the full image, preprocessing of the RPCA decomposition was performed for the forest image (P) to be processed, in which P = L + S. The position of the nonzero element (or one with a relatively larger absolute value) in S (sparse matrix) is extremely likely and corresponds to the location of the NGL in the original forest image. To enhance the NGL area in the original forest image, we proposed adding sparse matrix S to the original image P to obtain the enhanced image E = P + S. This step is called sparse enhancement (SE), as shown in Figure 9. This method was proposed in Reference [78] and was combined with the network architecture of deep learning for the first time in this paper. The original image was enhanced by sparse enhancement. The fundamental purpose of SE was to enhance the probable NGL area in the original forest image P to obtain the enhanced image E = P + S. The deep learning network was trained based on the enhanced image (Output P + S in Figure 9) and the corresponding ground truth.

According to the results of Reference [78], the enhancement effect of linear combination maintains the original spatial information, and it is enough to enhance the portion of NGL. Therefore, this paper uses the P+S image of the original image plus the sparse image for detection. As RPCA has different statistical methods, and according to the characteristics of different robust statistics, the calculated effects will also be different. We have used RPCA with different kernels such as the GA kernel (Grassmann Averages) [93], OPRMF kernel (Online Probabilistic Approach to Robust Matrix) [94], flip-SPCP-max-QN kernel (stable principal component pursuit) [95], and GoDec (Go Decomposition) [96], to find kernels with characteristics matching the NGL characteristics to enhance the accuracy of NGL detection.

### 2.6. Hybrid Convolutional Neural Network with Sparse Enhancement

The NGL image was processed by SE, and the sparse image in the original image was enhanced to highlight the NGL part in the image. Afterward, the enhanced NGL image was split into training and test datasets. As this paper had two images (Area 1 and Area 2), when Area 1 was used as training data, Area 2 would be used as test data. When Area 2 was used as training data, Area 1 would be used as test data.

#### 2.6.1. Training Data Set

The training images were generated by sliding windows, and the window size was 260 × 200. To increase the training data volume, the original image was segmented and slid rightwards for only 65 pixels (red window moving rightwards to the cyan window in Figure 10). When it slid to the edge, the window would move down 50 pixels and then slide to the left edge. These actions were repeated until the complete image is captured. There were 289 images split from the image as training data, and each image was 260 × 200. Figure 11 shows the training data generation of Area 1 and Area 2.

#### 2.6.2. Testing Data Set

The sliding window split method is also used for the testing data, and the window size was 200 × 260. Differing from the training data which slid rightwards only for 65 pixels, the test data slid rightwards 260 pixels at a time to avoid image overlap during the test (the red window moving to the cyan window in Figure 11). When the window slid to the edge, it would move down 200 pixels and then return to the left edge. The aforesaid action was repeated until the complete image was captured. There were 25 images split from the test data, and each image was 260 × 200. Figure 11 shows the test data splitting flowchart of Area 1 and Area 2.

After the training data and test data were split, model training was performed, as shown in Figure 12. First of all, the training data and ground-truth were trained using the model. The trained model predicted the test data and the performance was evaluated with the ground-truth of the testing data.

### 2.7. Evaluation of Detection Results

Two methods for evaluating accuracy were used in this paper. The first one was receiver operating characteristic (ROC) [97] analysis, which has recently received considerable interest in a wide range of applications in signal processing and medical diagnosis. To evaluate the detection performance of a ROC analysis, which makes use of a curve plotted as a function of detection probability, PD or True Positive, TP versus false alarm probability, PF or False Positive, FP are commonly used to assess the effectiveness of a detector. As an alternative to the use of ROC curves, the area under the curve (AUC), which has been widely used in medical diagnosis, was also calculated by the area under the ROC curve. Another method Cohen’s kappa coefficient (Kappa) [98] can measure the consistency between two classes. In image processing, the ROC curve is used to measure the effectiveness of the detector. Cohen’s kappa is an algorithm that evaluates the results of binarization and calculates consistency.

#### 2.7.1. ROC Curve

Receiver operating characteristic (ROC) analysis has been widely used in signal processing and communications to assess the efficiency of a detector. Figure 13 shows schematic diagram of ROC curve. The main concept of ROC analysis is a binary classification model, which has two classes of output results, e.g., correct/incorrect, match/mismatch, target/non-target, etc. ROC analysis which makes use of a curve to plot detection power (PD), or true positive (TP) versus false alarm probability (PF), or false positive (PF) is a commonly used evaluation tool to assess the effectiveness of a detector. As an alternative to the use of ROC curves, the area under the curve (AUC) is another index for evaluating the performance of detectors.

True positive (TP) and false positive (FPR) are used as the evaluation values of target detection accuracy. The accuracy is defined as follows:(6)$$\mathrm{Accuracy}=\frac{\mathrm{TP}+\mathrm{TN}}{P+N}$$

#### 2.7.2. Cohen’s Kappa

Cohen’s kappa coefficient is a statistical evaluation method for measuring consistency between two classes. Cohen’s kappa evaluates the results of binarization and calculates consistency. It is calculated by an error matrix identical to ROC curve, as shown in Table 2.

Where $P_{a}$ is the true value of NGL and what is detected is NGL, $P_{b}$ is the true value of NGL but what is detected is not NGL, $P_{c}$ is not the true value of NGL but what is detected is NGL, $P_{d}$ is not the true value of NGL and what is detected is not NGL. Cohen’s kappa can be defined in the following:(7)$$P_{o}=\frac{P_{a}+P_{d}}{P_{a}+P_{b}+P_{c}+P_{d}}=\frac{P_{a}+P_{d}}{N}$$
(8)$$P_{e}=P_{Yes}+P_{No}=\frac{P_{a}+P_{b}}{N}.\frac{P_{a}+P_{c}}{N}+\frac{P_{c}+P_{d}}{N}.\frac{P_{b}+P_{d}}{N}$$
(9)$$K=\frac{P_{o}-P_{e}}{1-P_{e}}=1-\frac{1-P_{o}}{1-P_{e}}$$

Coefficient K, as calculated by Cohen’s kappa is −1~1; if the detection results are completely correct, the K value is equal to 1; if K is equal to 0, the detection result is all NGLs or all non-NGL.

## 3. Results

This section is divided into three stages to compare the related experimental results. Section 3.1 performs a comparative analysis of the proposed parameters of BCE after the parameters are selected. The SE generated by RPCA with different kernels is tested in Section 3.2. Finally, Section 3.3 performs a comparative analysis of all the proposed CNNs.

### 3.1. BCE Parameter

This section discusses the data difference in the extended model proposed in this paper when the hyperparameter β of Loss function BCE is different, and the performance of each model is calculated and evaluated. This paper used ROC, ACC, and Kappa as the criteria of the evaluation score, because the effects of hyperparameter β on AUC and Kappa were compared. Figure 14 and Figure 15 only show the data of the flip-SPCP-max-QN kernel corresponding to different hyperparameter of β. According to the data drawing list of Area 1, the models had better performance when β=0.999, and various models had smaller differences when β=0.995, but the Kappa is lower. The performance of β=0.99 in AUC was only a few percentage points different from β=0.999, but the Kappa had a good performance in the two areas; therefore, β=0.99 was selected as the parameter in subsequent experiments.

### 3.2. Results of Different Kernels in SE

This section combines the SE preprocessed by RPCA with different kernels with the SegNet and U-Net of the BCE lost function and the proposed models used in the previous sections using Area 1 and Area 2. To review the differences among different kernels after preprocessing, the data in the previous section were sequenced according to AUC, true positive rate (TPR), FPR, and Kappa to compile a line chart, as shown in Figure 16 and Figure 17. According to the data drawing list of Area 1, the flip-SPCP-max-QN had better performance in detection (TPR) but had a larger increase in the amplitude of the misrecognition rate (FPR); therefore, its Kappa value was influenced greatly. With the GoDec method, the false alarm rate is relatively low, but the detection rate was also relatively low. To sum up, OPRMF had a relatively average performance in detection. In terms of Area 2, the GoDec kernel could not inhibit the background well, and the other three kernels had similar detection effects.

### 3.3. Results of Hybrid CNN Models

This section shows the comprehensive comparative results of various deep learning models using Area 1 and Area 2. This paper used the ROC, ACC, and Kappa as the criteria of the evaluation score. Table 3 shows the detection results of the original SegNet, U-Net, and our proposed model of Area 1 without SE. AUC, TPR, FPR, ACC, and Kappa were used for various evaluations. To compare with prior studies, in References [4,76,78], the global and local target detection results including Adaptive Coherence Estimator (ACE), Target Constrained Interference Minimized Filter (TCIMF), Constrained Energy Minimization (CEM), Subset CEM, Sliding Window-CEM (SW CEM), Adaptive Sliding Window-CEM (ASW CEM), and Weighted Background Suppression (WBS) version of above detectors were used for comparison. The results of target detection methods had better performance in AUC and TPR, but our proposed deep learning models especially in 3L-USN had the lowest FPR, and the highest ACC and Kappa were used up to 0.981 and 0.741, respectively.

Table 4 shows the detection results of all methods with SE. The SE shown in Table 4 used the flip-SPCP-max-QN kernel with its higher accuracy, but the loss function only used the BCE method (β = 0.99). Again, the results of target detection methods had better performance in AUC and TPR, but our 3L-USN had the lowest FPR, and the highest ACC and Kappa were used up to 0.98 and 0.743, respectively.

Figure 18 plots the ROC curves on the results in Table 3 and Table 4. Since target detection algorithms have very similar results, we only plot WBS-TCIMF, Subset WBS-CEM, and SW WBS-CEM. In addition, deep learning models provide hard decisions. In this case, their ROC curves are rectangles.

Table 5 shows the detection results of global and local target detection, SegNet, U-Net, and the proposed models of Area 2. In Area 2, the Kappa of U-Net was 0.692, which was the best among various models but provides the worst AUC, and TPR. Local target detection of Subset CEM has the best AUC but Kappa is not acceptable. However, our proposed 3L-SN and 3L-USN provide second-highest ACC and Kappa, slightly lower AUC but overall performance is better than others.

Table 6 shows the detection results of all methods in Area 2 with Sparse enhancement. In Area 2, again, the Kappa of U-Net was 0.708, which was the best among various models but provide the worst AUC and TPR. 3L-USN provides second-highest ACC and Kappa, slightly lower AUC but overall performance is better than others.

Figure 19 plots the ROC curves on the results in Table 4 and Table 5. To further highlight the detection results, Figure 20 and Figure 21 use three colors to mark false information (blue), hits (red), and target misses (yellow).

As the training sample, Area 1 had a smaller forest coverage, and only the right half had coverage. The detection result of Area 2 was worse than that of Area 1.

It is obvious our proposed deep learning models in Figure 21j–o have very few blue points (false alarm), especially in the right lower corner. According to the above resulting images and data, the original SegNet had excessive layers of pooling and the feature information of tiny targets had disappeared, which influenced the detection rate. As our hybrid CNN architecture reduced the pooling degree, the NGL detection rate was increased greatly. Figure 22 and Figure 23 show the data comparative histograms of SegNet, U-Net, and the proposed models of Area 1 and Area 2.

## 4. Discussion

Many intriguing findings can be observed from Table 3 and Table 4. Firstly, prior studies [4,76,78] used hyperspectral target detection algorithms. An advantage of hyperspectral target detection algorithms is they only require one spectral signature (target of interest) as an input parameter, while other prior knowledge such as multiple targets of interest or background is not required. However, the biggest problem is that these algorithms only provide soft decisions. As thresholding is required for the final decision, selecting a good threshold is a big challenge. This paper used deep learning models, which represent hard decisions without selecting thresholds. Moreover, global and local target detection methods are filter-based algorithms that apply a finite impulse response (FIR) filter to pass through the target of interest while minimizing and suppressing the background using a specific constraint [99]. It can detect many similar targets but also generates a very high false alarm rate. If the detection power or true positive rate (TPR) is the priority without considering a false alarm, using target detection algorithms is the appropriate manner. However, our proposed deep learning models provide a little lower AUC and TPR but achieve the highest overall accuracy and Kappa with the lowest false positive rate. Moreover, target detection methods are very sensitive since it only needs one input signature. Deep learning methods have more stable results but require a certain amount of training samples. Therefore, target detection and deep learning methods start from different points of view which can be applied in different circumstances. If you have limited information and concern computing time, matched filter-based target detection can meet demands. On the other hand, if you already have a certain amount of information and computing time is not your major concern, deep learning methods can fulfill your needs.

Secondly, our proposed models had much better results than the original SegNet and U-Net, meaning the information of tiny target NGL could be maintained by reducing the pooling degree, and U-SegNet with increased skip connection could enhance the spatial information of the sample. The performance in AUC, ACC, and Kappa were much better than the original SegNet and U-Net, especially in Kappa, which was up to 0.74.

Thirdly, NGL can be characterized by four unique features. First, they have unexpected presence. Secondly, they have a very low probability of occurrence, Thirdly, they have a relatively small sample population. Last but most important, their signatures are distinct from their surrounding pixels. As the features of NGL are coincident with the characteristics of sparsity, the NGL signal can be enhanced by our proposed sparse enhancement (SE) technique even areas may differ in the appearance of NGL. Table 7 and Table 8 list different models after SE. The enhanced models were enhanced in AUC and TPR. According to the SegNet model result in Area 1, the AUC of the enhanced image was greatly increased by 0.56 and the TPR was increased by 0.13, but there were some false alarms; therefore the Kappa decreased slightly. To sum up the results, the overall NGL enhancement was effective and the AUC and TPR of each model of Area 1 were increased significantly. Last but not the least, according to the data comparison table of Area 2 shown in Table 8, the AUC and TPR of various models were reduced but the ACC and Kappa were increased significantly. It could be observed in Figure 18 that the blue false alarm area in the right lower corner after SE was reduced greatly; therefore, the performance in ACC and Kappa was upgraded as the FPR decreased greatly. The AUC and TPR were reduced slightly but remained acceptable; therefore, the overall performance of SE in Area 1 indicated that the detection rate was increased, and the false alarm was reduced effectively in Area 2. Generally, the Kappa was improved significantly. As the NGL accounted for only 3% of the full image, the Kappa was relatively objective for evaluation. Another reason was that Area 1 is used as a training sample for testing Area 2. Area 1 has less NGL coverage indicates fewer training samples and Area 2 includes more species of trees in the right lower corner; therefore, the overall performance in Area 2 was worse than that in Area 1. To sum up the above data, the SE preprocessing method had a better detection effect on various models.

## 5. Conclusions

The use of telemetry to monitor the health of forest ecosystems has an important influence on controlling global warming. Prior studies have investigated and proposed target detection algorithms for NGL detection. It can achieve high detection power but cause a very high false alarm rate as well and only provide a soft decision, not a hard decision. To address the above issues, this paper makes several contributions. First and foremost, this paper applied deep learning in the SegNet network architecture and proposed four models, 3LSN, 3L-USN, 2L-USN, and 2L-Conv-USN by decreasing the number of pooling and adding skip connection (SC) technique. The original SegNet model has a high pooling frequency that can cause the low-level features of tiny targets to blur or even disappear; therefore, we reduced the number of layers of pooling. On the one hand, the model could be downsized; on the other hand, the loss of tiny targets could be reduced while maintaining information to increase the detection performance. When the skip connection (SC) of U-Net was used, the spatial information of the same level was connected up through up-sampling on the bottom layer. In comparison to SegNet and U-Net, the four models had better performance in NGL detection of tiny targets and training time. Secondly, this paper used the weighted BCE loss function since NGL in the image belongs to imbalanced data; the weight of NGL can be increased, and the weight of the background was suppressed at the same time. Third, the SE technique was used as preprocessing to enhance NGL signal before entering deep learning models. According to the experimental results, after Area 1 was preprocessed by SE, the NGL detection rate was increased successfully, and the number of false alarms was reduced greatly in Area 2. Last but not the least, this paper conducts a complete comparative analysis, analyzing the pros and cons of global/local target detection algorithms and deep learning methods. To sum up, this paper is believed to be the first work that highlighted the use of hybrid deep learning models for the detection of NGL. The experimental results demonstrate the overall performance of our proposed models outperforms local/global target detection algorithms and original state of art deep learning models.

## Figures and Tables

**Figure 1 sensors-21-02077-f001:**
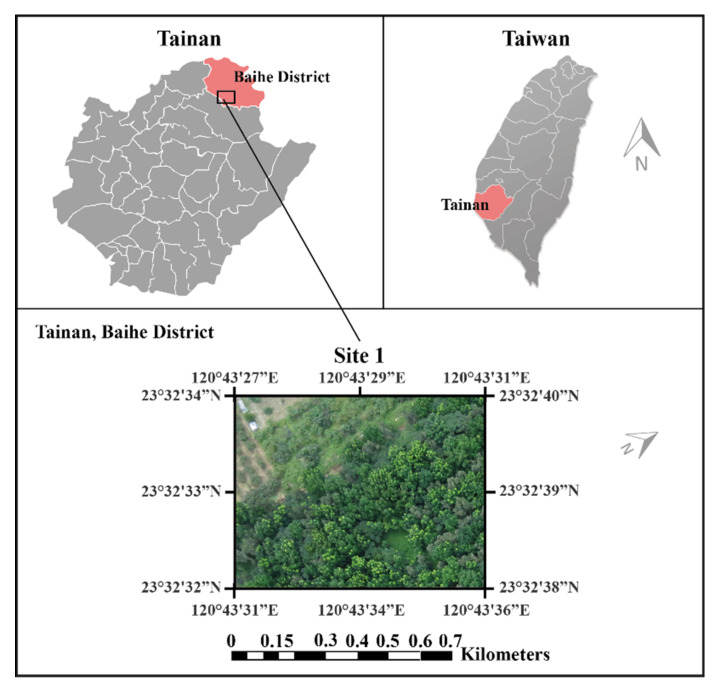
Study site.

**Figure 2 sensors-21-02077-f002:**
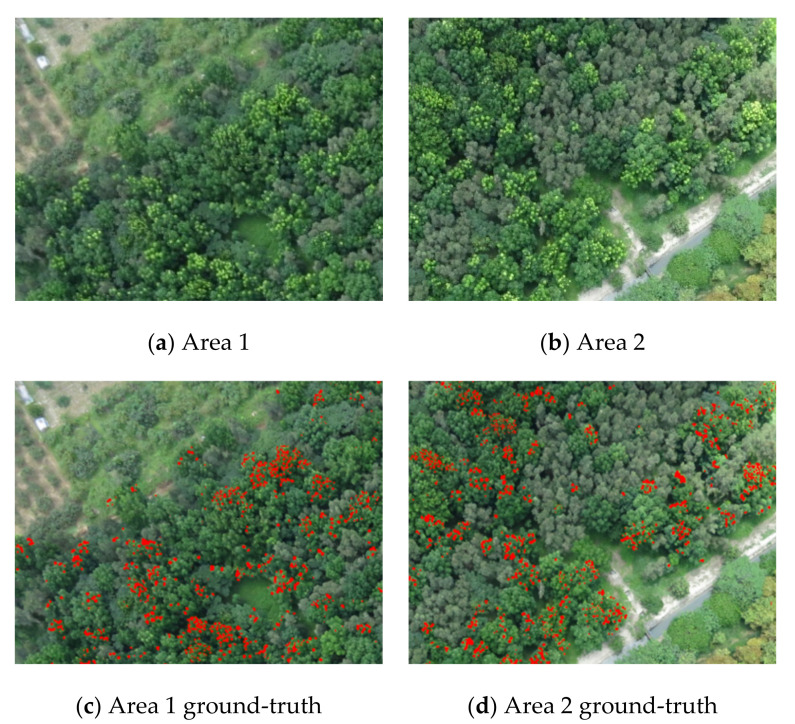
The original images and the corresponding Ground-truths of Area 1 and Area 2.

**Figure 3 sensors-21-02077-f003:**
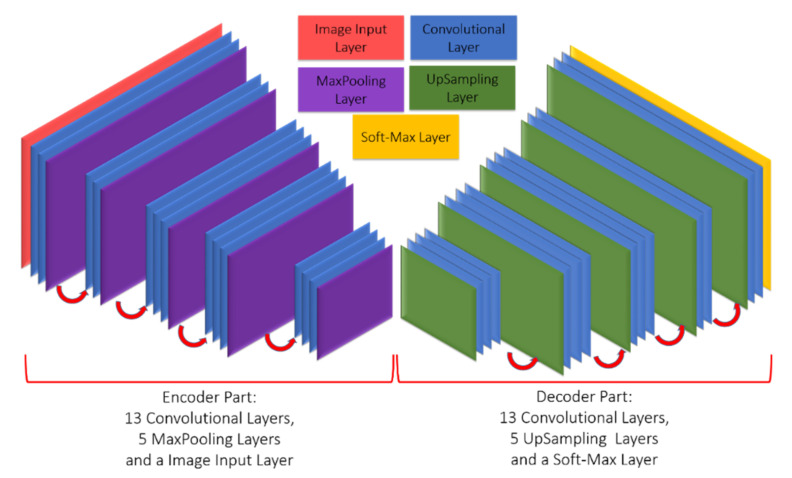
SegNet architecture.

**Figure 4 sensors-21-02077-f004:**
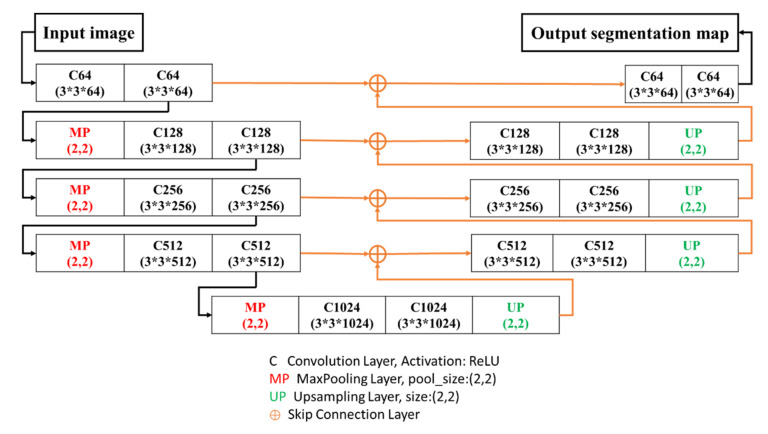
U-Net architecture.

**Figure 5 sensors-21-02077-f005:**
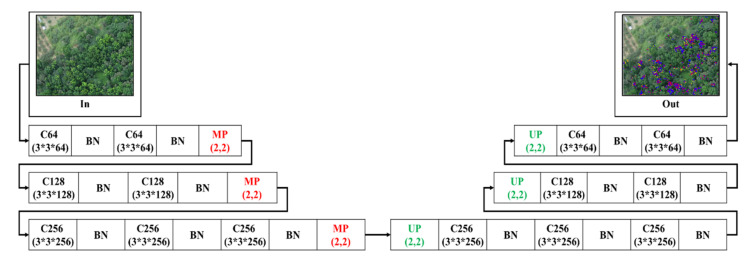
3-Layer SegNet architecture.

**Figure 6 sensors-21-02077-f006:**
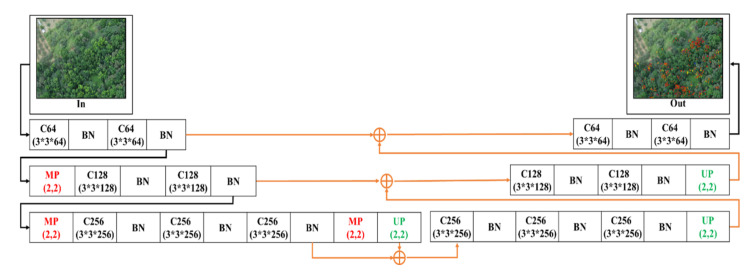
3-Layer skip connection (SC) SegNet architecture.

**Figure 7 sensors-21-02077-f007:**
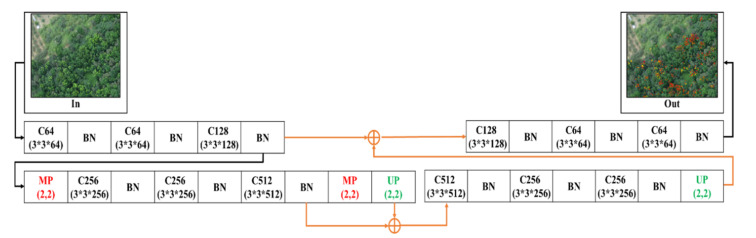
2-Layer SC SegNet architecture.

**Figure 8 sensors-21-02077-f008:**
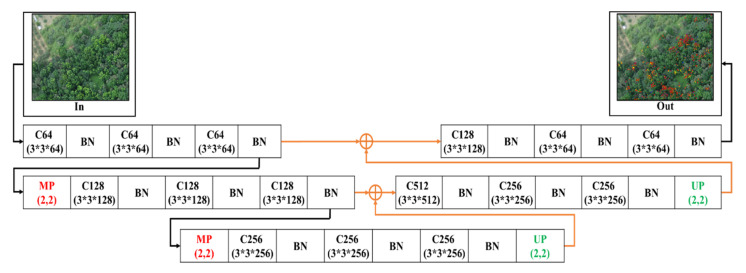
2-Layer Conv-U-SegNet architecture.

**Figure 9 sensors-21-02077-f009:**
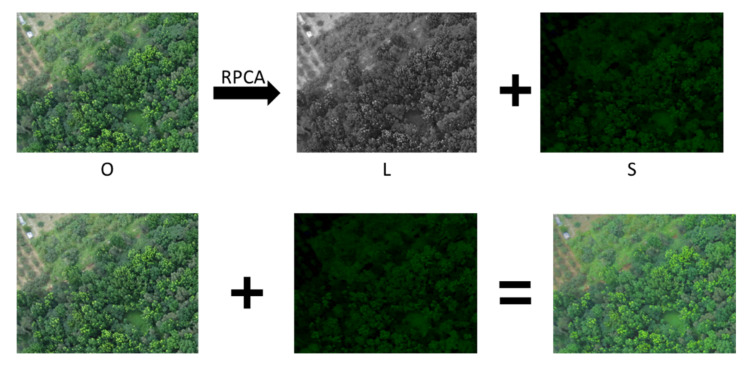
Sparse enhancement flowchart.

**Figure 10 sensors-21-02077-f010:**
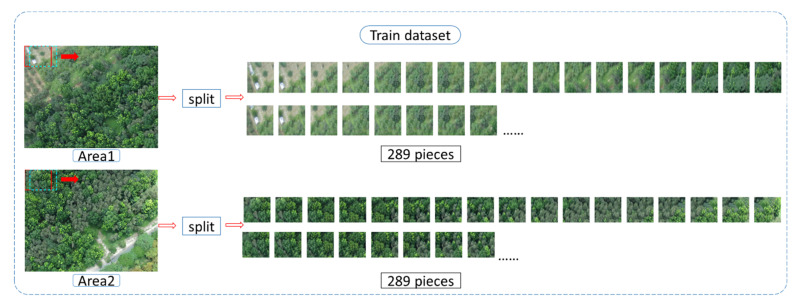
Training data process.

**Figure 11 sensors-21-02077-f011:**
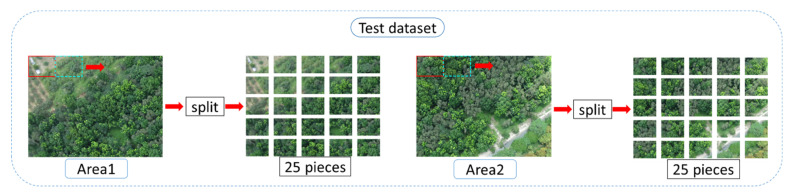
Data testing process.

**Figure 12 sensors-21-02077-f012:**
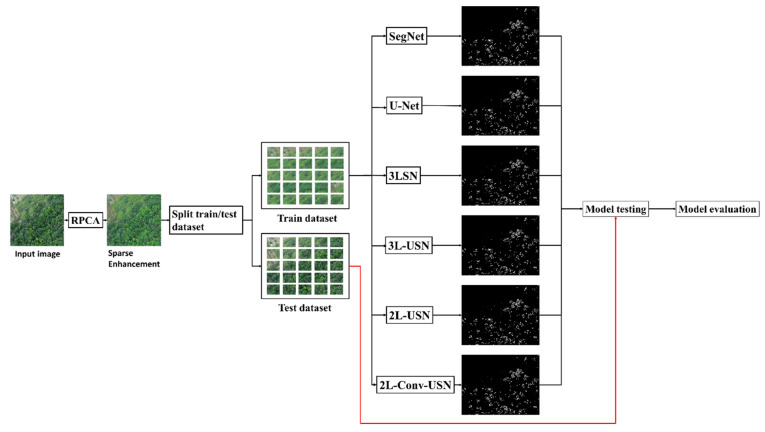
Flowchart of the hybrid convolutional neural network with the sparse enhancement technique.

**Figure 13 sensors-21-02077-f013:**
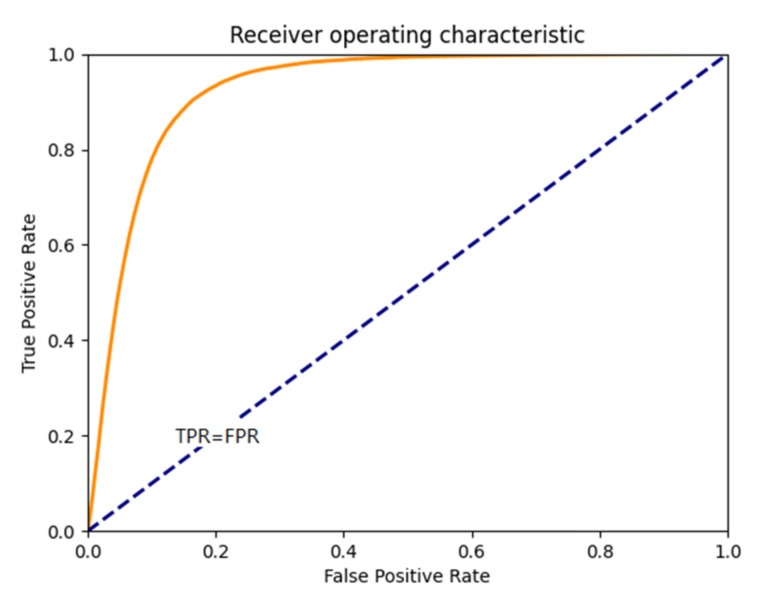
Receiver operating characteristic (ROC) curve.

**Figure 14 sensors-21-02077-f014:**
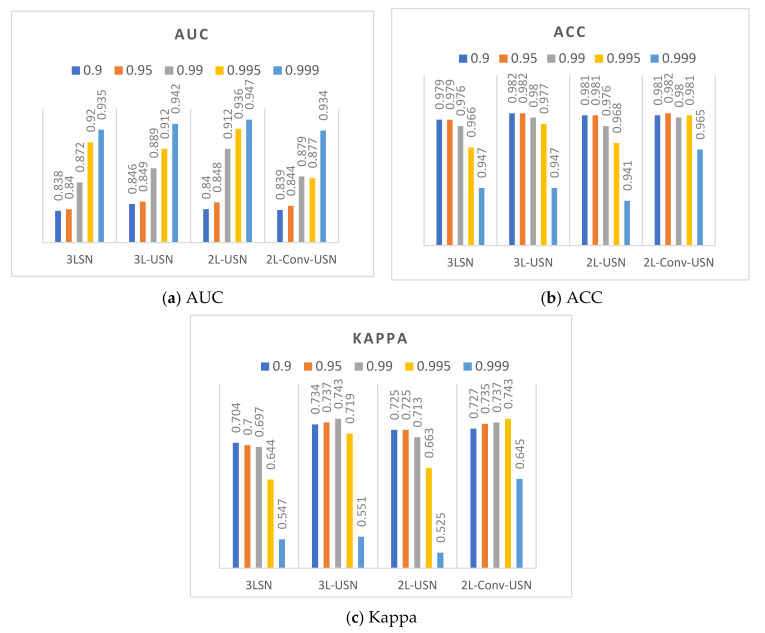
Data bar chart of Area 1 using five hyperparameters of β and four models.

**Figure 15 sensors-21-02077-f015:**
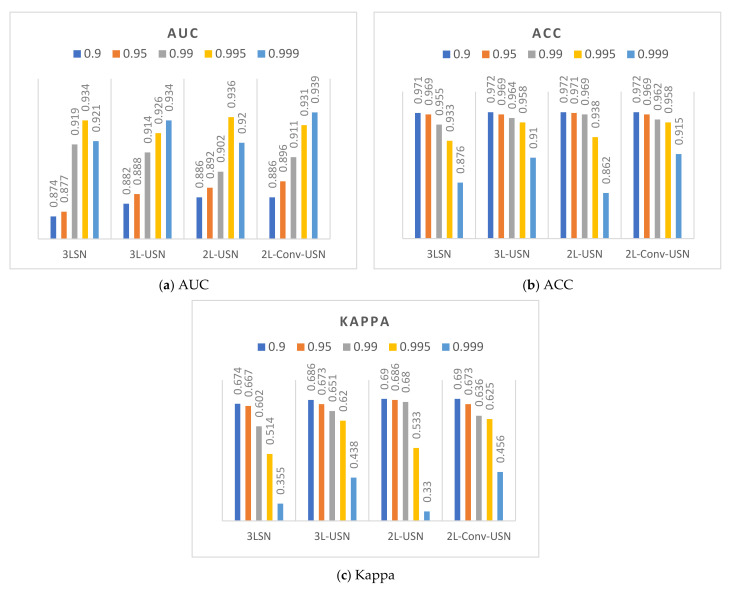
Data bar chart of Area 2 using five hyperparameters of β and four models.

**Figure 16 sensors-21-02077-f016:**
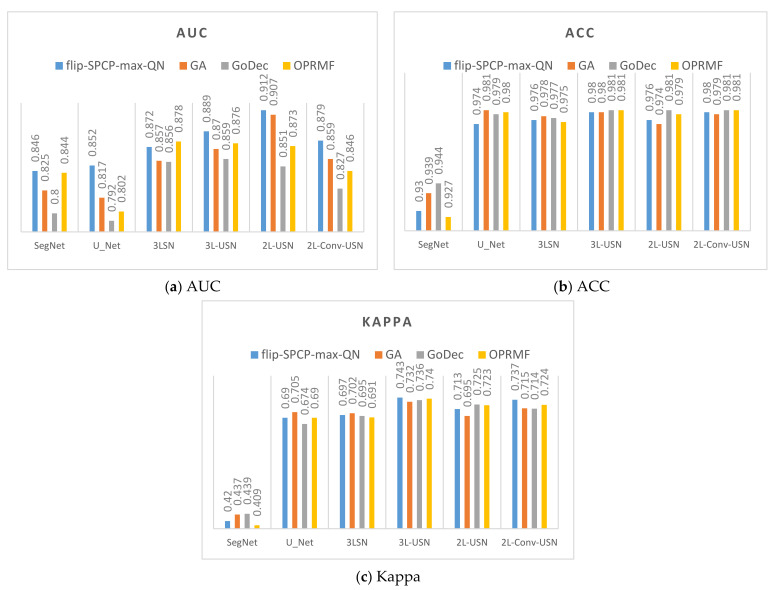
Data bar chart of the area under the curve (AUC) and Kappa of Area 1 using four kernels + four models.

**Figure 17 sensors-21-02077-f017:**
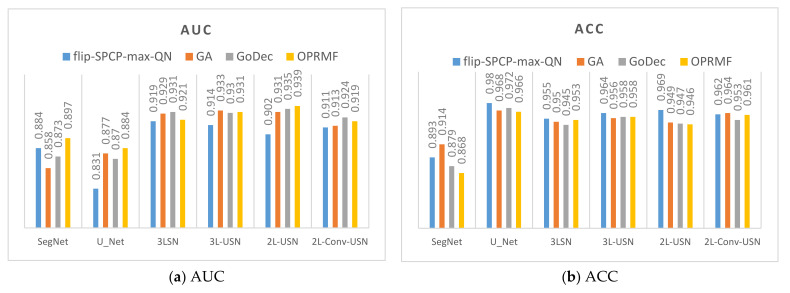
Data bar chart of the AUC and Kappa of Area 2 using four kernels + four models.

**Figure 18 sensors-21-02077-f018:**
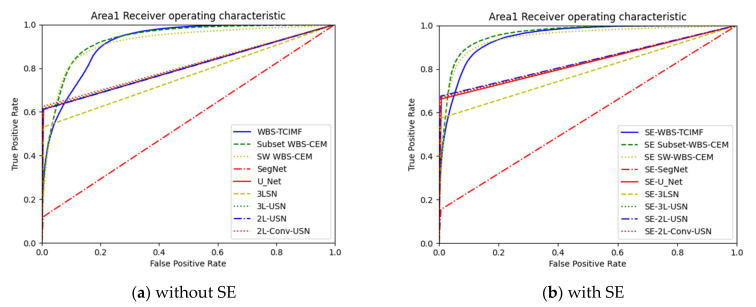
ROC curve of the global and local target detection results, SegNet, U-Net, and our proposed deep learning mod-els in Area 1.

**Figure 19 sensors-21-02077-f019:**
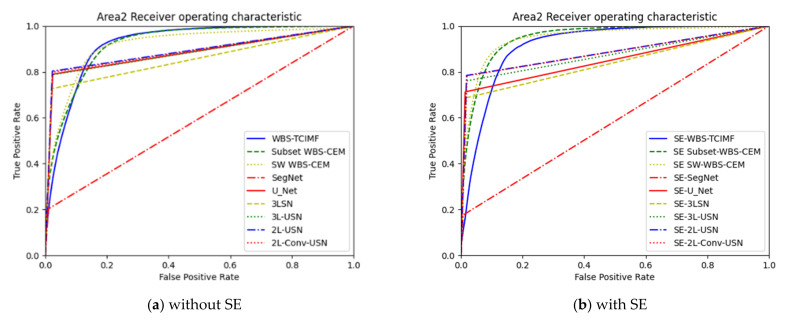
ROC curve of the global and local target detection results, SegNet, U-Net, and our proposed deep learning models in Area 2.

**Figure 20 sensors-21-02077-f020:**
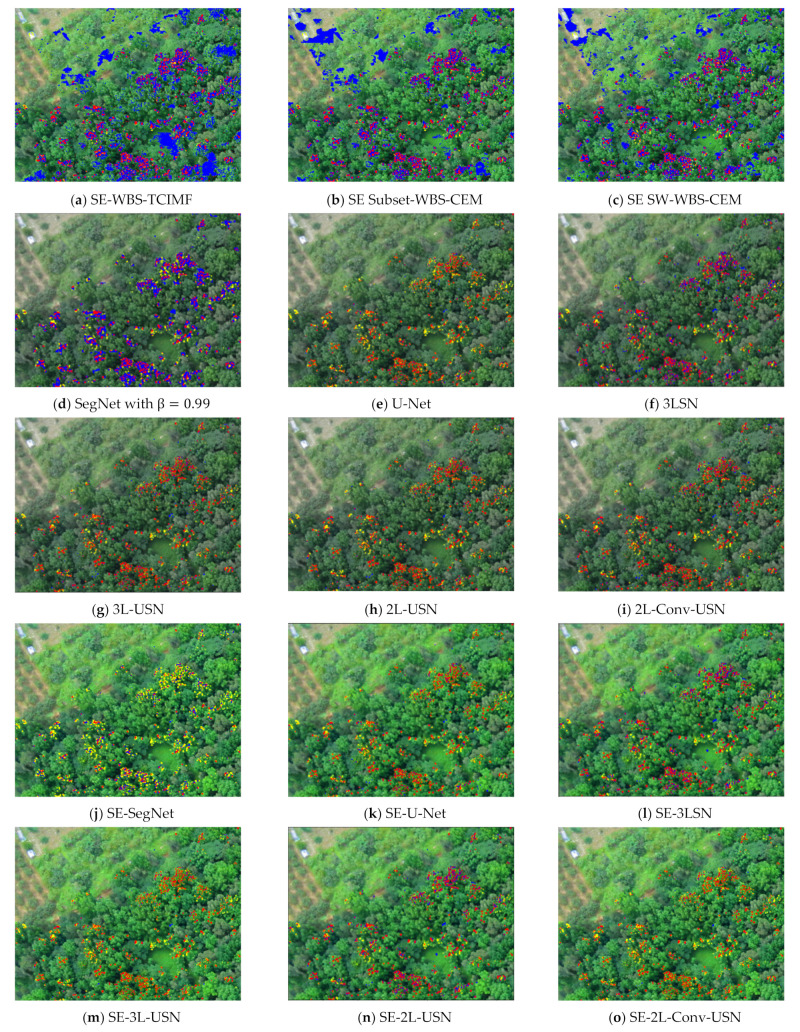
Resulting images of the global and local target detection results, SegNet, U-Net, and our proposed deep learning models in Area 1.

**Figure 21 sensors-21-02077-f021:**
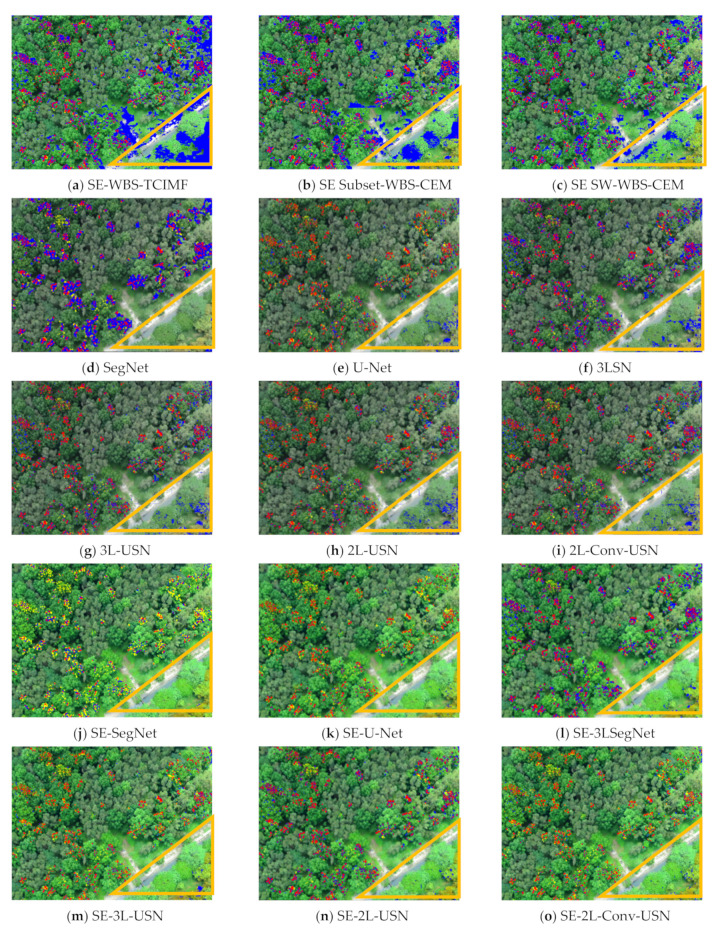
Resulting images of the global and local target detection results, SegNet, U-Net, and our proposed deep learning models in Area 2.

**Figure 22 sensors-21-02077-f022:**
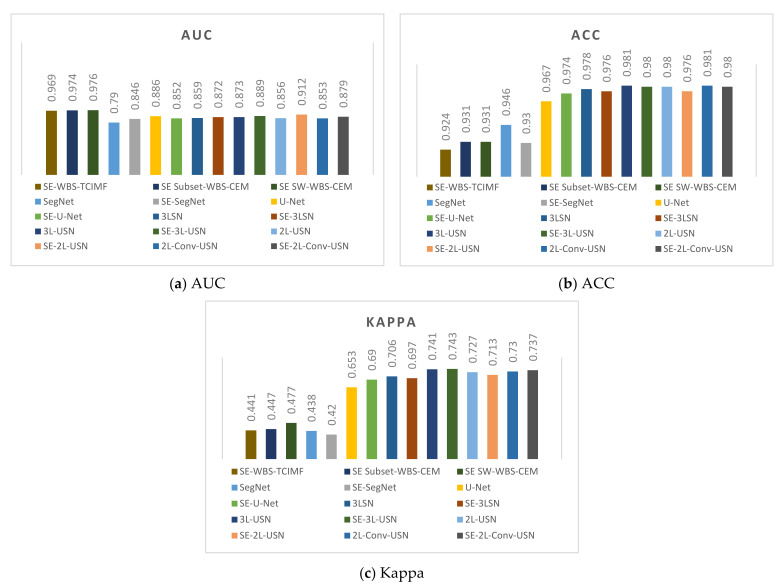
Model data comparative histogram of Area 1.

**Figure 23 sensors-21-02077-f023:**
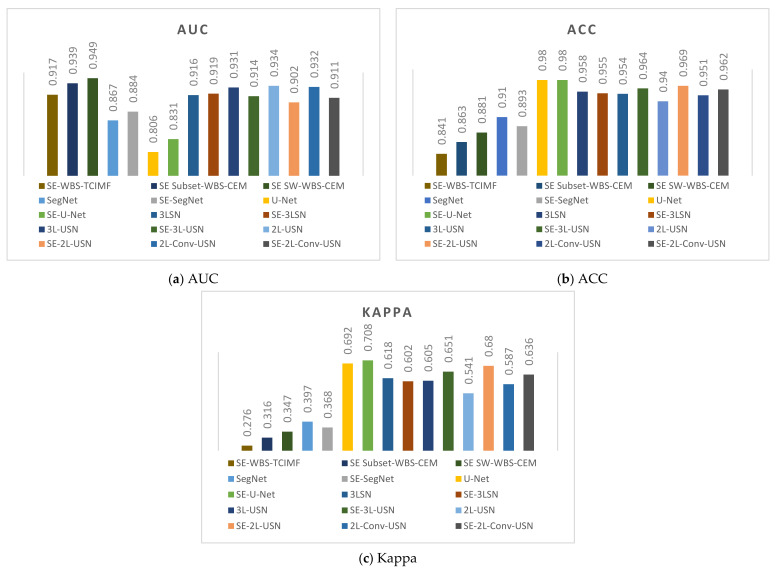
Model data comparative histogram of Area 2.

**Table 1 sensors-21-02077-t001:** Model code description.

Code	Code Description	Parameter Information
**C**	Convolution Layer	Kernel Size: (3, 3) Padding: same Activation: ReLU Filter Size -> C64, C128, C256
**BN**	BatchNormalization Layer	
**MP**	MaxPooling Layer	
**UP**	UpSampling Layer	
**Out**	Convolution Layer	Kernel Size: (1, 1) Padding: same Activation: sigmoid Filter Size: 1
**⊕**	Skip Connection Layer	

**Table 2 sensors-21-02077-t002:** Error matrix of Cohen’s kappa.

Error Matrix		Ground Truth		Total
		NGL (p)	Non-NGL (n)	
Detection	NGL (p′)	True Positive $P_{a}$	False Positive $P_{b}$	$P_{a}+P_{b}$
	Non-NGL (n′)	False Negative $P_{c}$	True Negative $P_{d}$	$P_{c}+P_{d}$
Total		$P_{a}+P_{c}$.	$P_{b}+P_{d}$	Total Pixels N

**Table 3 sensors-21-02077-t003:** Detection comparison among global, local target detection, SegNet, U-Net, and our proposed deep learning models of Area 1 without sparse enhancement (SE).

**Global Target Detection**					
	**AUC**	**PD/TPR**	**PF/FPR**	**ACC**	**Kappa**
Traditional ACE	0.9417	0.8827	0.1099	0.8897	0.3389
WBS-ACE (ED)	0.9375	0.8697	0.1113	0.888	0.3313
Traditional TCIMF	0.95737	0.8903	0.1076	0.8924	0.3472
WBS-TCIMF(ED)	0.9583	0.8906	0.1082	0.8917	0.3458
Traditional CEM	0.9606	0.8915	0.1022	0.8976	0.3604
WBS-CEM(ED)	0.9713	0.9029	0.0738	0.9254	0.4483
Sparse WCEM	0.971	0.9067	0.08	0.9194	0.4288
**Local Target Detection**					
	**AUC**	**PD/TPR**	**PF/FPR**	**ACC**	**Kappa**
Subset CEM	0.9685	0.9099	0.0874	0.9125	0.4074
Subset WBS-CEM	0.9725	0.9177	0.0827	0.9173	0.4248
SW CEM	0.9694	0.9151	0.0811	0.9188	0.4289
SW WBS-CEM	0.9749	0.9204	0.0729	0.9269	0.4588
ASW CEM	0.9704	0.9257	0.0753	0.9248	0.4526
ASW WBS-CEM	0.9781	0.9325	0.0673	0.9327	0.4847
**Deep Learning Based (Proposed)**					
**Model**	**AUC**	**TPR**	**FPR**	**ACC**	**Kappa**
SegNet	0.79	0.621	0.042	0.946	0.438
U-Net	0.886	0.799	0.026	0.967	0.653
3LSN	0.859	0.73	0.012	0.978	0.706
3L-USN	0.773	0.757	0.01	0.981	0.741
2L-USN	0.856	0.821	0.009	0.98	0.727
2L-Conv-USN	0.853	0.715	0.009	0.981	0.73

The red/green values indicate the best/worst performance in each evaluation metric.

**Table 4 sensors-21-02077-t004:** Detection comparison among the global and local target detection results, SegNet, U-Net, and our proposed deep learning models of Area 1 with SE.

**Global Target Detection**					
	**AUC**	**PD/TPR**	**PF/FPR**	**ACC**	**Kappa**
SE-WBS-ACE	0.9658	0.9154	0.0887	0.9115	0.4058
SE-WBS-TCIMF	0.9691	0.8963	0.0749	0.9241	0.4417
SE-WBS-CEM	0.9713	0.9029	0.0738	0.9254	0.4483
Sparse WCEM	0.971	0.9067	0.08	0.9194	0.4288
**Local Target Detection**					
	**AUC**	**PD/TPR**	**PF/FPR**	**ACC**	**Kappa**
SE Subset-WBS-CEM	0.9749	0.9159	0.0681	0.9313	0.4744
SE SW-WBS-CEM	0.9766	0.9241	0.0681	0.9316	0.4778
SE ASW-WBS-CEM	0.9796	0.9324	0.0616	0.9382	0.5076
**Deep Learning Based (Proposed)**					
**Model**	**AUC**	**TPR**	**FPR**	**ACC**	**Kappa**
SE-SegNet	0.846	0.751	0.063	0.93	0.42
SE-U-Net	0.852	0.719	0.014	0.974	0.69
SE-3L SegNet	0.872	0.76	0.015	0.976	0.697
SE-3L-U-SegNet	0.889	0.789	0.012	0.98	0.743
SE-2L-U-SegNet	0.912	0.844	0.019	0.976	0.713
SE-2L-Conv-U-SegNet	0.879	0.77	0.011	0.98	0.737

The red/green values indicate the best/worst performance in each evaluation metric.

**Table 5 sensors-21-02077-t005:** Detection comparison among the global and local target detection results, SegNet, U-Net, and our proposed deep learning models of Area 2 without SE.

**Global Target Detection**					
	**AUC**	**TPR**	**FPR**	**ACC**	**Kappa**
Traditional ACE	0.943	0.899	0.131	0.871	0.325
WBS-ACE	0.948	0.910	0.140	0.862	0.312
Traditional TCIMF	0.952	0.907	0.138	0.864	0.315
WBS-TCIMF	0.888	0.925	0.230	0.777	0.202
Traditional CEM	0.950	0.903	0.132	0.870	0.324
WBS-CEM	0.893	0.925	0.220	0.787	0.212
**Local Target Detection**					
	**AUC**	**TPR**	**FPR**	**ACC**	**Kappa**
Subset CEM	0.952	0.893	0.114	0.887	0.359
Subset WBS-CEM	0.917	0.923	0.180	0.824	0.255
SW CEM	0.921	0.849	0.110	0.889	0.351
SW WBS-CEM	0.923	0.903	0.146	0.856	0.299
ASW CEM	0.942	0.875	0.097	0.901	0.390
ASW WBS-CEM	0.932	0.918	0.141	0.862	0.313
**Deep Learning Based (Proposed)**					
	**AUC**	**TPR**	**FPR**	**ACC**	**Kappa**
SegNet	0.867	0.819	0.086	0.91	0.397
U-Net	0.806	0.617	0.006	0.98	0.692
3L SegNet	0.916	0.87	0.038	0.958	0.618
3L-USN	0.931	0.906	0.044	0.954	0.605
2L-USN	0.934	0.928	0.059	0.94	0.541
2L-Conv-USN	0.932	0.912	0.048	0.951	0.587

The red/green values indicate the best/worst performance in each evaluation metric.

**Table 6 sensors-21-02077-t006:** Detection comparison among the global and local target detection results, SegNet, U-Net, and our proposed deep learning models of Area 2 with SE.

**Global Target Detection**					
	**AUC**	**TPR**	**FPR**	**ACC**	**Kappa**
SE-WBS-ACE	0.950	0.911	0.126	0.876	0.339
SE-WBS-TCIMF	0.917	0.909	0.162	0.841	0.276
SE-WBS-CEM	0.916	0.916	0.172	0.832	0.265
**Local Target Detection**					
	**AUC**	**TPR**	**FPR**	**ACC**	**Kappa**
Subset SE-WBS-CEM	0.939	0.921	0.140	0.863	0.316
SW SE-WBS-CEM	0.938	0.898	0.120	0.881	0.347
ASW SE-WBS-CEM	0.949	0.924	0.117	0.885	0.363
**Deep Learning Based (Proposed)**					
**Model**	**AUC**	**TPR**	**FPR**	**ACC**	**Kappa**
SegNet	0.884	0.874	0.106	0.893	0.368
U-Net	0.831	0.669	0.008	0.98	0.708
3LSN	0.919	0.881	0.042	0.955	0.602
3L-USN	0.914	0.86	0.032	0.964	0.651
2L-USN	0.902	0.828	0.024	0.969	0.68
2L-Conv-USN	0.911	0.855	0.033	0.962	0.636

The red/green values indicate the best/worst performance in each evaluation metric.

**Table 7 sensors-21-02077-t007:** Area 1 experiment data comparison table.

Area	Model	AUC	TPR	FPR	ACC	Kappa
**1**	**SegNet**	+0.560 ▲	+0.130 ▲	+0.021 ▲	−0.016 ▼	−0.018 ▼
	**U-Net**	−0.034 ▼	− 0.080 ▼	− 0.080 ▼	+0.008 ▲	+0.037 ▲
	**3LSN**	+0.013 ▲	+0.030 ▲	+0.003 ▲	−0.002 ▼	−0.009 ▼
	**3L-USN**	+0.114 ▲	+0.032 ▲	+0.032 ▲	−0.001 ▼	+0.002 ▲
	**2L-USN**	+0.056 ▲	+0.013 ▲	+0.010 ▲	−0.004 ▼	−0.014 ▼
	**2L-Conv-USN**	+0.026 ▲	+0.055 ▲	+0.002 ▲	−0.001 ▼	+0.007 ▲

**Table 8 sensors-21-02077-t008:** Area 2 experiment data comparison table.

Area	Model	AUC	TPR	FPR	ACC	Kappa
**2**	**SegNet**	+0.017 ▲	+0.055 ▲	+0.020 ▲	−0.017 ▼	−0.029 ▼
	**U-Net**	+0.025 ▲	+0.052 ▲	+0.002 ▲	0.000 ▲	+0.016 ▲
	**3LSN**	+0.003 ▲	+0.011 ▲	+0.004 ▲	−0.003 ▼	−0.016 ▼
	**3L-USN**	−0.017 ▼	−0.046 ▼	−0.012 ▼	+0.010 ▲	+0.001 ▲
	**2L-USN**	−0.032 ▼	−0.100 ▼	−0.035 ▼	+0.029 ▲	+0.141 ▲
	**2L-Conv-USN**	−0.021 ▼	−0.057 ▼	−0.015 ▼	+0.011 ▲	+0.049 ▲

The red/green values indicate the best/worst performance in each evaluation metric.

## Data Availability

Data and source codes are available from the authors upon reasonable request.

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
