# Peer review of "Hybrid Deep Learning Models with Sparse Enhancement Technique for Detection of Newly Grown Tree Leaves"

_sensors, 2021, doi:10.3390/s21062077_

Round 1
Reviewer 1 Report
In this study, Chen et al. created a hybrid deep learning model to classify newly growth tree leaves. The contribution is that they manually collected the data and assessed different deep learning models to learn that data. However, there are many major points that need a lot of efforts to improve:
1. English language writing was not acceptable in the current form. The authors should re-check and revise carefully.
2. There are missing a lot of literature review related to this problem (deep learning-based growth tree leaves detection)
3. How did the authors label the ground truths is a big question. How many people perform this step? How to verify the correctness?
4. The authors did not mention clearly about the datasets. How much data did they use? How to separate training/testing?
5. Which evaluation method did they use in this study?
6. How did the authors deal with hyperparameter tuning? It is an important process for deep learning implementation.
7. There are figures (i.e., Figure 4) is a copy from other sources without any reference. Thus the authors should re-plot or cite the original source to avoid replication.
8. The authors used ROC for analysis, but they did not show any ROC curve.
9. Deep learning has been used in previously published works i.e., PMID: 33260643 and PMID: 31362508. Therefore, the authors are suggested to refer to more works to attract a broader readership.
10. As I have known, the authors have sealed with the segmentation problem also. Thus why did they not use some common metrics in this problem i.e. Dice, IoU, ...? I think ROC is for binary classification, how did they only measure the predictive performance with ROC & Kappa?
11. The authors should have some external validation data.
12. The "Discussion" section should be rephrased. Currently, the authors used bullet points to discuss, it is not encouraged in science.
13. The authors should compare the predictive performance to previous works on the same problem.
14. Source codes should be released for replicating the study.
15. Quality of figures should be improved significantly.
16. There must have space before the reference number.
Author Response
Responses to Reviewer 1’s comments:
1) English language writing was not acceptable in the current form. The authors should re-check and revise carefully.
Response:
Thanks for the comments. Our manuscript has re-checked by a native English speaking colleague and used a professional English editing service.
2) There are missing a lot of literature review related to this problem (deep learning-based growth tree leaves detection).
Response:
Thanks for the comment. We added reference about research of leaves using deep learning in the introduction part, but most of those are highlighted in leaf recognition [44], plant classification [45-46], leaf disease classification and detection [47-52]. Plant and leaf identification [53-55] using deep learning is a relatively new field. [44] used a basic and Pre-trained GoogleNet for leaf recognition. [53] applied a CNN model to identify leaf veins. [54] proposed a weed segmentation system by combing the segmentation algorithm and CNN models. [55] proposed LeafNet, a CNN-based plant identification system. The LeafNet architecture is also similar to well-known AlexNet and CifarNet. According to the literature review, there is rare research investigates the effect of the segmentation of hybrid deep learning manners for detection of newly grown leaf (NGL) in detail. We believe our study first attempted to examine detection of newly grown leaf (NGL) using deep learning technique in high resolution UAV images. We added the above review in the Section 1.
3) How did the authors label the ground truths is a big question. How many people perform this step? How to verify the correctness?
Response:
Reviewer’s comments are well-taken. The authors have deployed a few permanent plots over the broadleaf forest for research of forest growth [82-83] in 2008, where a series of ground inventory is annually conducted for stand dynamics [84]. The data have been successfully derived forest canopy height model [85] and have investigated the feasibility of global/local target detection algorithms for the detection of NGL by our prior studies [4,78,80]. As shown in Figure 2 (c-d), the NGLs can be visually inspected by experts due to their appearance of being bright and light green and amassed over tree crowns. According to a row of several years of inventory, the ground truth of the NGL over the images were visually interpreted by two professors from the Department of Forestry at National Chiayi University, Taiwan and also validated in situ. We added the above explanations in Section 2.1.
4) The authors did not mention clearly about the datasets. How much data did they use? How to separate training/testing?
Response:
Thanks for the comment. We added 2.1-2.2 to describe our data set and the ground truth. Since we only had two images (Area 1 and Area 2), when Area 1 was used as training data, Area 2 would be used as testing data. When Area 2 was used as training data, Area 1 would be used as testing data. The detailed information is mentioned in 2.6.1-2.6.2.
5) Which evaluation method did they use in this study?
Response:
Thanks for the comments. We added evaluation methods in 2.7.1-2.7.2. .
6) How did the authors deal with hyperparameter tuning? It is an important process for deep learning implementation.
Response:
Thanks for the comments. The parameters of the deep learning we used follow the original U-Net including kernel sizes and filter sizes, however we applied Balanced cross-entropy (BCE) as the loss function to deal with the imbalanced data problem. The β of BCE is based on try and error. The detailed experimental results can be found in section 3.1.
- There are figures (i.e., Figure 4) is a copy from other sources without any reference. Thus the authors should re-plot or cite the original source to avoid replication.
Response:
Thanks for the comments. We re-plotted all figures.
- The authors used ROC for analysis, but they did not show any ROC curve.
Response:
Thanks for the comments. We added the ROC curves in Figures 18-19.
- Deep learning has been used in previously published works i.e., PMID: 33260643 and PMID: 31362508. Therefore, the authors are suggested to refer to more works to attract a broader readership.
Response:
Reviewer’s comments are well-taken. We referred more works as the reviewer suggested in the introduction part.
- As I have known, the authors have sealed with the segmentation problem also. Thus why did they not use some common metrics in this problem i.e. Dice, IoU, ...? I think ROC is for binary classification, how did they only measure the predictive performance with ROC & Kappa?
Response:
Thanks for the comments. Since our prior studies [4,78,80] make use of target detection point of view to detect NGL, in order to compare the results, we follow the same evaluation methods which are ROC and Kappa. In addition, target detection algorithms in [4,78,80] only provide soft decision. ROC is an appropriate method to evaluate the performance. This paper started with the segmentation of deep learning application which achieves pixel-level prediction and each pixel is classified according to their category, dividing image into foreground and background which provide the same purpose of binary classification and target detection as previous studies.
- The authors should have some external validation data.
Response:
Thanks for the comments. Since we only have two images (one testing data and one training data), there is no more data can be used for external validation. It will be our future work once we collect more UAV data set.
- The "Discussion" section should be rephrased. Currently, the authors used bullet points to discuss, it is not encouraged in science.
Response:
Reviewer’s comments are well-taken. The discussion part is rephrased as the reviewer suggested.
- The authors should compare the predictive performance to previous works on the same problem.
Response:
Reviewer’s comments are well-taken. We compare the results of our previous works including global/local target detection algorithms with our proposed works in the Section 3.3. We also discuss the pros and cons of all algorithms in the Section 4.
- Source codes should be released for replicating the study.
Response:
Reviewer’s comments are well-taken. Data and source codes are available from the authors upon reasonable request. Here is the download link: https://www.asuswebstorage.com/navigate/a/#/s/8FAEEA6A4639474E83BA43C1B195C149Y
- Quality of figures should be improved significantly.
Response:
Reviewer’s comments are well-taken. We updated all of the figures with better quality.
- There must have space before the reference number.
Response:
Thanks for the comments. We made the correction.
Reviewer 2 Report
1.The authors used methods, tools and models from other researchers. In the data collection drones are used introduced by others. The several segnet models, are taken from other researchers. The choice of many parameters is taken from others without further motivation. Sofar the paper is about an application of deep learning models Segnet. Model improvements , adaptation are not clearly discussed.
2. The goal of the study is not well defined. Globally the paper is about the detection of area of sprouts. But areas may differ in the appearance of sprouts, density, distribution etc.
3. The photo taken by the drone are not validated. Is it possible to inspect the pictures by human experts in the field. The pictures are taken on 12 july, is that a representative data for new sprout detection? At the end of page 3 the authors discuss distribution maps as ground truth. It is not clear how these maps are used. It is for us not clear how the raw pictures can be used as input, is it vaidated that sprouts are detectable.
4. Our main problem is that the results are not benchmarked with other methods or applications.
Author Response
Responses to Reviewer 2’s comments:
1) The authors used methods, tools and models from other researchers. In the data collection drones are used introduced by others. The several segnet models, are taken from other researchers. The choice of many parameters is taken from others without further motivation. So far the paper is about an application of deep learning models Segnet Model improvements, adaptation are not clearly discussed.
Response:
Reviewer’s comments are well-taken. We revised the introduction and discussion part. The data collection is done by our group. The current data image took us more than one year to complete all the tasks including flying done, validating ground truth and filed inventory. Our previous studies [4,78,80] used the same data set to investigate the global/local target detection algorithms for detection of newly grown leaves (NGL). Differing from prior studies in terms of target detection point of view, this paper started with the segmentation of deep learning application which achieves pixel-level prediction and each pixel is classified according to their category, dividing image into foreground and background which provide the same purpose of binary classification and target detection as previous studies. This paper combined hybrid models with U-NET and SegNet, reducing the number of pooling layers to keep the information of NGL and using skip connection to extract feature information from low-level information in the hopes of enhancing NGL features. In addition, this paper used sparse enhancement preprocessing before the network architecture of deep learning. As the area of NGL to be detected in the forest image had sparse-like characteristics compared with the full image, the NGL in the original image were enhanced by this method. Moreover, the choice of parameters is designed by the features of NGL. This paper increases the weight of NGL by using the weighted BCE loss function since NGL in the image belongs imbalanced data.
2) The goal of the study is not well defined. Globally the paper is about the detection of area of sprouts. But areas may differ in the appearance of sprouts, density, distribution etc.
Response:
Thanks for the comments. The goal of this paper is to investigate the development of robust deep learning-based algorithms to diagnose the growth of trees, and even climatic change by detecting newly grown leaves (NGL) from high resolution UAV images and the performance of using sparse enhancement (SE) technique as preprocessing in our proposed models. In addition, NGL can be characterized by four unique features. First, they have unexpected presence. Secondly, they have very low probability of occurrence, Thirdly, they have relatively small sample population. Last but most important, their signatures are distinct from its surrounding pixels. As the features of NGL are coincident with the characteristics of sparsity, the NGL signal can be enhanced by our proposed sparse enhancement (SE) technique even areas may differ in the appearance of NGL. We added the above explanations in the Section 4. .
3) The photo taken by the drone are not validated. Is it possible to inspect the pictures by human experts in the field. The pictures are taken on 12 july, is that a representative data for new sprout detection? At the end of page 3 the authors discuss distribution maps as ground truth. It is not clear how these maps are used. It is for us not clear how the raw pictures can be used as input, is it validated that sprouts are detectable.
Response:
Thanks for the comments. The authors have deployed a few permanent plots over the broadleaf forest for research of forest growth [82-83] in 2008, where a series of ground inventory is annually conducted for stand dynamics [84]. The data have been successfully derived forest canopy height model [85] and have investigated the feasibility of global/local target detection algorithms for the detection of NGL by our prior studies [4,78,80]. As shown in Figure 2 (c-d), the NGLs can be visually inspected by experts due to their appearance of being bright and light green and amassed over tree crowns. According to a row of several years of inventory, the ground truth of the NGL over the images were visually interpreted by two professors from the Department of Forestry at National Chiayi University, Taiwan and also validated in situ.
This paper is to develop an algorithm to detect newly grown leaves (NGL) not sprout. Leaf development is a process of dynamic plant growth responding to plant physiology and environmental signals [81]. Leaves start to develop from the apical meristems of branches. It is unachievable to visualize a newly sprouted leaflet from the UAV at initiation stage of the life cycle of a leaf. However, after a period of gradual development, the newly grown leaf is normally at a size of a centimeter and can be visualized from a distance. As a results, July is the good time to fly UAV to detect newly grown leaves in the south of Taiwan. We added the above explanations in Section 2.1.
4) Our main problem is that the results are not benchmarked with other methods or applications.
Response:
Thanks for the comments. In order to compare with prior studies in [4,78,80], the global and local target detection results including Adaptive Coherence Estimator (ACE), Target Constrained Interference Minimized Filter (TCIMF), Constrained Energy Minimization (CEM), Subset CEM, Sliding Window-CEM (SW CEM), Adaptive Sliding Window-CEM (ASW CEM) and Weighted Background Suppression (WBS) version of above detectors in [4,78,80] were used for comparison. The results of target detection methods had better performance in AUC and TPR, but our proposed deep learning models especially in 3L-USN had lowest FPR, and highest ACC and Kappa were used up to 0.981 and 0.741 respectively. We added the comparative analysis in Section 3.3. In addition, global and local target detection methods are matched filter-based algorithms applied a finite impulse response (FIR) filter to pass through the target of interest, while minimizing and suppressing the background using a specific constraint [104]. It can detect as many as similar targets but also generates very high false alarm rate. However, our proposed deep learning models provide a little lower AUC and TPR but achieve highest overall accuracy and Kappa with lowest false positive rate. Moreover, target detection methods are very sensitive since it only needs one input signature. Deep learning methods have more stable results but require a certain amount of training samples. Therefore, target detection and deep learning methods start from different points of view which can be applied in different circumstance. If you have limited information and concern computing time, matched filter based target detection can meet demands. On the other hand, if you already have certain amount of information and computing time is not your major concern, deep learning methods can fulfill needs.
Secondly, our proposed models had much better results than original SegNet and U-Net, meaning the information of tiny target NGL could be maintained by reducing the pooling degree, and U-SegNet with skip connection could enhance the spatial information of the sample. The performance in AUC, ACC and Kappa was much better than original SegNet and U-Net, especially in Kappa, which was up to 0.74. We added the above explanations in the Section 4.
Reviewer 3 Report
This paper proposes a hybrid deep learning CNN model with sparse enhancement for detection of newly grown tree leaves, which is quite interesting and significant. It is well organized and carefully writtern. The model design is reasonable and the experimental results demonstrate its effectiveness. Therefore, I would like to recommend it to be accepted and published.
Author Response
This paper proposes a hybrid deep learning CNN model with sparse enhancement for detection of newly grown tree leaves, which is quite interesting and significant. It is well organized and carefully writtern. The model design is reasonable and the experimental results demonstrate its effectiveness. Therefore, I would like to recommend it to be accepted and published.
Response:
We appreciate reviewer’s excellent comments.
Round 2
Reviewer 1 Report
My previous comments have been addressed well.